# Semantic-Enhanced Prototypical Network for Universal Novel Category Discovery

## Abstract

We address the challenging task of Universal Novel Category Discovery (UniNCD) in image classification, where models must distinguish between common and novel categories while avoiding the misclassification of novel categories as private-known ones. Previous prototype-based approaches face two major challenges: first, they significantly increase the negative transfer risk by often misaligning novel categories with private-known categories; second, they lead to sub-optimal prototypes because traditional prototype learning ignores diverse object characteristics of images, resulting in insufficient semantic guidance when optimizing instance representations using only instance-level prototypical distributions. To tackle these challenges, we present a *Semantic-Enhanced Prototypical Network*, dubbed SEPNet. This prototypical network is enhanced by refined prototypes and enriched semantics to learn better representations and avoid negative transfer, including three key ideas: (1) we design a *Prototype Refinement* (PR) strategy that can decouple common, private-known, and novel categories from unlabeled data, which can exclude misaligned prototypes to avoid negative transfer; (2) we attach prototypical distribution to each patch of images, which embed enhanced semantic information to prototypes and guide prototypical contrastive learning and, (3) we design a *patch-entropy balance* (PEB) method to encourage sparser patch-level prototypical distributions while maintaining the uniformity of dense distributions, sparsity emphasizes dominant category characteristics, and uniformity avoids the misguidance of irrelevant disturbance, thereby enhancing the distinctiveness of instances to the prototypes. Our method demonstrates superior performance on the UniNCD task across three benchmark datasets, outperforming existing state-of-the-art approaches by approximately 3.4% in terms of accuracy. We will release our code for reproduction.

## 1 Introduction

### 1.1 Universal Novel Category Discovery

Recent image classification advancements often rely on predefined categories for training, but these methods fall short in real-world situations with emerging categories, leading to the Novel Category Discovery (NCD) task, where models adapt to novel categories from unlabeled images. Existing approaches assume either "Disjoint NCD" (Hsu et al., 2017; Han et al., 2019; 2020) or "Open-World NCD" settings (Vaze et al., 2022; Cao et al., 2021; Rizve et al., 2022), but practical scenarios often involve private-known categories in labeled data, which do not exist in the unlabeled data.

In this paper, we consider a more challenging task, Universal Novel Category Discovery (UniNCD). In UniNCD, the labeled dataset includes "private-known" categories that do not exist in the unlabeled dataset, and "common" categories are shared between labeled and unlabeled data, while the rest in the unlabeled dataset are "novel" categories. For example, a model trained on doves, quail (common categories), and swans (a private-known category) encounters ducks (a novel category) and should avoid misclassifying ducks as swans (Figure 1 left). In this context, we aim to classify unlabeled images as common or novel categories without misclassifying them as private-known categories.

Recent research in Open-world Novel Category Discovery focused on learning known and novel categories in a decoupled manner by aligning the prototypes of unlabeled data with those of labeled

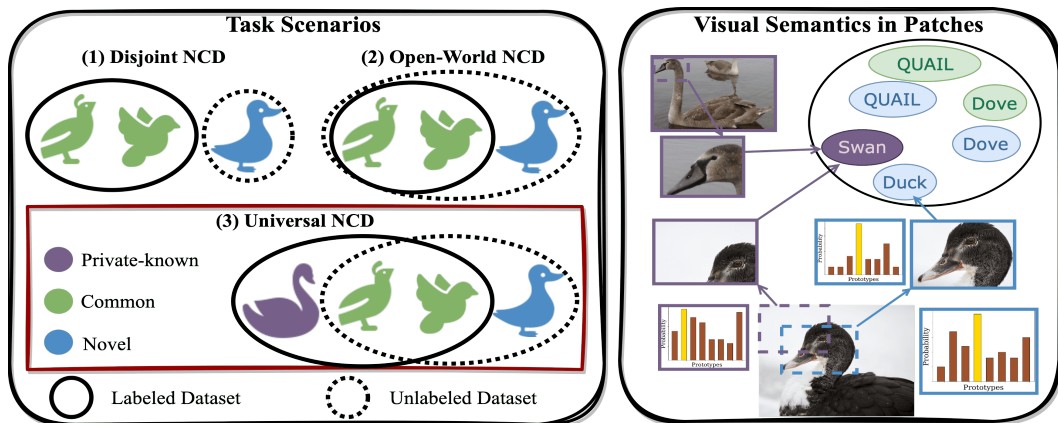

Figure 1: Left: task scenarios of Novel Category Discovery. Universal Novel Category Discovery introduces private-known categories (swan) in labeled data. Right: prototypical distributions of different patches of an image, the patches contain more discriminative semantic information, leading to sparser patch-level distributions (in green rectangle), while others are more general or irrelevant.

data (An et al., 2023). However, matching the entire labeled dataset with the unlabeled dataset can lead to prototype misalignment, that is to say, some novel prototypes in unlabeled data are incorrectly aligned with those of private-known categories. This misalignment significantly increases the *negative transfer risk* (NTR) (Wang et al., 2019), where the model misclassifies unlabeled data as private-known categories. Moreover, previous works also employed *Prototypical Contrastive Learning* (PCL) (Snell et al., 2017; Li et al., 2020), which performs iterative clustering and representation learning to learn discriminative representations for a group of similar instances (Yu et al., 2022; Zhao et al., 2023; An et al., 2023). However, these methods solely relied on instance-level prototypical distributions, failing to comprehensively consider the diverse characteristics within images. This limitation leads to insufficient semantic guidance during instance representation optimization, which, in turn, yields sub-optimal prototypes within prototypical networks. We hypothesize that using a single prototypical distribution to represent the classification probability of an image is similar to using a single word embedding to represent its complex conceptual details, which falls short in capturing multiple characteristics in complex images.

## 1.2 Our Solutions

To address the above challenges, we introduce a *Semantic-Enhanced Prototypical Network* (**SEPNet**). In this method, a prototypical network is refined by our *Prototypes Refinement* (PR) strategy and improved by enriched semantics with our *Patch-Entropy Balance* (PEB) method. These enhancements aim to improve representation learning and mitigate negative transfer risk (section 3). First, multi-task pre-training (Zhang et al., 2022) is combined with the generalized contrastive loss to enhance the model generalization ability. By leveraging instance-level contrastive learning that doesn't rely on known labels for supervision, this approach encourages the model to concentrate on the intrinsic semantic information of objects and gain a comprehensive insight into their diverse characteristics, bolstering the model's robustness to noise in the supervision signal.

Second, the PR strategy is employed during the initial training epochs to rectify prototype misalignment caused by private-known categories. This strategy leverages a Z-score filter along with robust rank statistics (Han et al., 2020) to access the prototype alignment confidence, which is often overlooked by previous methods(Zhao et al., 2023; An et al., 2023). By quantifying this confidence, misaligned prototypes can be identified and refined iteratively, thereby mitigating the NTR. Additionally, the PR strategy aids in estimating private-known categories and generating pseudo labels for unlabeled data, facilitating the creation of negative sets for PCL.

Third, we attach a prototypical distribution to each patch of images, creating patch-level prototypical distributions to guide PCL with enriched semantics. Notably, we observe certain regions with discriminative details have sparse patch-level prototypical distributions, while others with general or

irrelevant information exhibit denser distributions (Figure 1 right). Inspired by this, our PEB method promotes sparsity in sparse prototype distributions while maintaining uniformity in dense ones. This balancing encourages prototypes to capture distinctive category characteristics and reduces confusion from irrelevant noise, enhancing instance distinctiveness relative to the prototypes.

We evaluate our method on three public benchmarks (section 4) within the UniNCD task. Our method surpasses the performance of existing state-of-the-art techniques (section 2) by approximately 3.4%. The main contributions of our paper are summarized as follows:

1. We propose a *Semantic-Enhanced Prototypical Network* (SEPNet) for Universal Novel Category Discovery. SEPNet incorporates a *Prototype Refinement* (PR) strategy to effectively rectify prototype misalignment caused by private-known categories, significantly mitigating negative transfer risk.
2. We propose a *Patch-Entropy Balance* (PEB) method to enrich semantics by attaching a prototypical distribution to each image patch, creating patch-level prototypical distributions. PEB method is employed to promote sparsity in sparse prototype distributions while maintaining uniformity in dense ones, encouraging prototypes to capture distinctive category characteristics while reducing confusion from irrelevant noise.
3. Experimental results demonstrate the superiority of SEPNet, outperforming existing state-of-the-art techniques in UniNCD tasks across three datasets.

## 2 RELATED WORK

### 2.1 NOVEL CATEGORY DISCOVERY (NCD)

In novel category discovery (Hsu et al., 2017; Han et al., 2019; 2020; Zhong et al., 2021; Fini et al., 2021; Li et al., 2023), the goal is to cluster the unlabeled dataset, comprised of disjoint but similar categories, utilizing knowledge from the labeled set. However, this task tends to overlook known categories in the unlabeled dataset, assuming all categories in it are novel. Unlike conventional NCD, Open-World NCD assumes the unlabeled set also shares all categories in the labeled set, making model creation challenging (Vaze et al., 2022; Sun & Li, 2022; Rizve et al., 2022; Cao et al., 2021; Liu et al., 2023; Zhang et al., 2023). Pu et al. (2023) introduces a semi-supervised Gaussian Mixture Model with dynamic prototype selection and prototypical contrastive learning for representation improvement, iteratively refining clusters for unlabeled data. An et al. (2023) addresses Open-world NCD by effectively separating known and novel categories, explicit category-specific knowledge transfer, and enhanced feature learning via Semantic-aware Prototypical Learning. However, in UniNCD, the division of the label space is stricter, which emphasizes the influence of private-known categories in the labeled dataset, which simulates more realistic real-world scenarios.

### 2.2 UNIVERSAL DOMAIN ADAPTION (UNIDA)

Universal Domain Adaptation (UniDA) addresses category-shift issues similar to UniNCD. Previous works such as Universal Adaption Network (You et al., 2019), DANCE (Saito et al., 2020), and OVANet (Saito & Saenko, 2021) use confidence scores to recognize novel categories. However, they classify all novel categories as one, limiting their utility for classification and NCD tasks. Additionally, they don't leverage prior information from labeled data to guide the categorization and clustering of unlabeled data. In this study, we extend the universal setting to NCD.

## 3 METHOD

### 3.1 PRELIMINARY

**Notation.** In UniNCD task, the training dataset $\mathbb{D}_{tr}$ consists of a labeled dataset $\mathbb{D}_l = \{(\mathbf{x}_i^l, y_i^l)\}_{i=1}^{N_l}$ with known categories $\mathbb{C}_l$, and an unlabeled dataset $\mathbb{D}_u = \{(\mathbf{x}_i^u, y_i^u)\}_{i=1}^{N_u}$ with certain known categories plus its novel categories, collectively referred to as $\mathbb{C}_u$. The test dataset $\mathbb{D}_{ts}$ includes both known and novel categories. For clarity, three finer-grained concepts are defined: common categories, private-known categories, and novel categories denoted as $\mathbb{C}_{com}, \mathbb{C}_{pk}, \mathbb{C}_n$, where $\mathbb{C}_{com} = \mathbb{C}_l \cap \mathbb{C}_u$, $\mathbb{C}_{pk} = \mathbb{C}_l \backslash \mathbb{C}_{com}$ and $C_n = \mathbb{C}_u \backslash \mathbb{C}_{com}$. The common categories are the set of categories shared by

both datasets, while private-known categories exclusively exist in the labeled dataset. The model is trained and optimized on $\mathbb{D}_{tr}$ and then evaluated on $\mathbb{D}_u$ and $\mathbb{D}_{ts}$. Given an unlabeled instance $\mathbf{x}_u$, the goal of the UniNCD task is to employ an encoder $\phi : \mathcal{X} \to \mathbb{R}^d$ that encodes $\mathbf{x}_u$ to a feature embedding $\mathbf{f} = \phi(\mathbf{x})$ with dimension $d$, then either classify it as one of the common categories $\mathbb{C}_{com}$ or group it with similar unlabeled instances to form one of the discovered novel categories amongst $\mathbb{C}_n$. Crucially, the model must refrain from misclassifying it into private-known categories $\mathbb{C}_{pk}$.

**Overall Process.** Our approach has three principal stages: (1) A transformer-based model is pre-trained with loss displayed as Eq. 1, which is composed of the supervised contrastive loss $\mathcal{L}_{\text{SCL}}$ on public datasets plus the labeled internal dataset (Khosla et al., 2020), and the self-supervised contrastive loss $\mathcal{L}_{\text{SSCL}}$ on the unlabeled internal dataset (Chen et al., 2020), resulting a pre-trained model $\phi'$; (2) Then a prototypes alignment and refinement strategy is applied to iteratively update the prototypes of labeled and unlabeled data $\{\boldsymbol{\mu}^l, \boldsymbol{\mu}^u\}$, along with the estimation of prototypes representing private-known categories $\boldsymbol{\mu}^l_{pk}$ and pseudo labels for unlabeled data $\{\hat{y}^u_i\}_{i=1}^{||\mathbb{D}_u||_1}$; (3) Afterward, a model is trained using prototypical contrastive loss on instance level, with an entropy-regularized uniformity loss on patch level. The overall process is also depicted in Appendix B.

$$\mathcal{L}_{\text{MTP}} = \mathcal{L}_{\text{SCL}}(\mathbb{D}_l; \boldsymbol{\theta}) + \mathcal{L}_{\text{SSCL}}(\mathbb{D}_u; \boldsymbol{\theta}) \tag{1}$$

**Prototype Alignment.** Taking unlabeled data as examples, we maintain a feature memory, denoted as $\mathbf{V}^u$, to store their features. Each $\mathbf{v}^u_i$ represents the feature vector of $\mathbf{x}^u_i$. The memory is initialized with $\phi'(\mathbf{x}^u_i)$ and updated using momentum $m$ after each batch (Eq. 2a). Then $K$-means clustering is performed on $\mathbf{V}^u$ to obtain unlabeled clusters $\boldsymbol{\Lambda}^u$ and then the unlabeled data prototype matrix $\boldsymbol{H}_u = \begin{bmatrix} \boldsymbol{\mu}^u_1 & \cdots & \boldsymbol{\mu}^u_{||\mathbb{C}_u||_1} \end{bmatrix}$ (Eq.2b). The labeled data prototypes matrix $\boldsymbol{H}_l$ is computed using the ground truth, and memory updating for labeled data is similar to unlabeled data.

$$\mathbf{v}^u_i \leftarrow m\mathbf{v}^u_i + (1 - m)\phi(\mathbf{x}^u_i) \tag{2a}$$

$$\boldsymbol{\mu}^u_j = \frac{1}{||\boldsymbol{\Lambda}^u_j||_1} \sum_{\mathbf{v}^u_i \in \Lambda^u_j} \mathbf{v}^u_i \tag{2b}$$

Aiming to separate known and novel categories within unlabeled data, we adopt a bipartite matching strategy following An et al. (2023). This strategy aligns prototypes $\{\boldsymbol{\mu}^u_j\}_{j=1}^{||\mathbb{C}_u||_1}$ and $\{\boldsymbol{\mu}^l_j\}_{j=1}^{||\mathbb{C}_l||_1}$, by seeking an optimal permutation denoted as $\hat{\mathcal{P}}$. This permutation minimizes the Euclidean distance between $\{\boldsymbol{\mu}^l_j\}_{j=1}^{||\mathbb{C}_l||_1}$ and their corresponding permutations in $\{\boldsymbol{\mu}^u_j\}_{j=1}^{||\mathbb{C}_u||_1}$, as defined by the equation:

$$\hat{\mathcal{P}} = \arg\min_{\mathcal{P} \in \mathcal{P}_{all}} \sum_{i=1}^{||\mathbb{C}_l||_1} (||\boldsymbol{\mu}^l_i - \boldsymbol{\mu}^u_{\mathcal{P}(i)}||) \tag{3}$$

Once the optimal matching is achieved, the categories represented by unlabeled data prototypes are considered to be those of their matched labeled data prototypes, with indexes denoted as $\{\hat{\mathcal{P}}(i)\}_{i=1}^{||\mathbb{C}_u||_1}$.

## 3.2 PROTOTYPES REFINEMENT

However, in UniNCD, directly aligning unlabeled data prototypes with labeled data prototypes can significantly increase NTR, because prototypes representing novel categories may be misaligned with those representing private-known categories, namely, misaligned prototypes. To address this, we introduce a distance-based Prototype Refinement strategy to decouple prototypes into three groups, $\boldsymbol{\mu}_{pk}, \boldsymbol{\mu}_{com}, \boldsymbol{\mu}_n$, representing private-known, common, and novel categories, respectively (Figure 2 middle). We presume that the distance within misaligned prototypes is apparently larger than others (Figure 2 left), and calculate the Z-score for distances within aligned prototypes (Eq. 4) to filter out these misaligned prototypes with a high NTR.

$$Z_i = \frac{d(\boldsymbol{\mu}^l_i, \boldsymbol{\mu}^u_{\hat{\mathcal{P}}(i)}) - \mu}{\sigma} \tag{4}$$

Here, $d(\cdot, \cdot)$ represents the Euclidean distance, while $\mu$ and $\sigma$ represent the mean and standard deviation of all distances, respectively. Aligned prototypes with Z-scores above the 0.75-quantile of all Z-scores are considered misaligned prototypes and removed from $\hat{\mathcal{P}}$ ($q$ is set to 0.75 for best

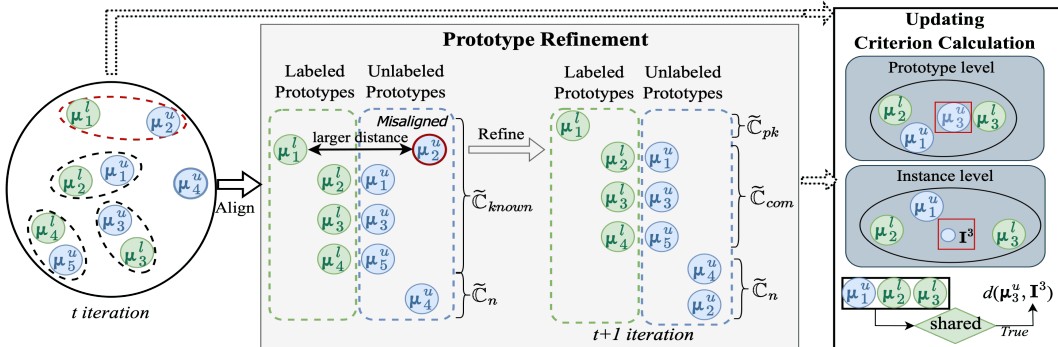

Figure 2: **Prototype Refinement (PR)**. Prototypes are decoupled into those representing private-known, common, and novel categories, then unlabeled data prototypes are updated with RS criteria.

performance, the choice of $q$ is discussed in Appendix H.2). Next, we order the unlabeled data prototypes according to categories represented by their corresponding labeled data prototypes, and the remaining unlabeled data prototypes represent novel categories.

During early training, instability in $K$-Means and indiscriminate representations can result in poor-quality prototypes, which can negatively impact the performance of our model. To address this issue, we propose an iterative updating strategy that combines intra-cluster distance and robust rank statistics (Figure 2 right). This strategy aims to enhance the quality of unlabeled data prototypes by iteratively refining them, especially during early training. Specifically, for each unlabeled data prototype ($\boldsymbol{\mu}_i^u$) and its related instances ($\mathbb{I}_i$), we select instances that share the same $k$ nearest prototypes of $\boldsymbol{\mu}_i^u$ to calculate the updating criterion $d^{rs}$ as follows:

$$\mathbb{I}_i^{rs} = \{\mathbf{l}_j^i \mid \mathbf{l}_j^i \in \mathbb{I}_i, \text{ where } \boldsymbol{\mu}_i^u \text{ and } \mathbf{l}_j^i \text{ share the same } k \text{ nearest prototypes}\}$$

$$d^{rs} = \sum_i^{||\mathbb{C}_u||_1} \frac{1}{||\mathbb{I}_i^{rs}||_1} \sum_{\mathbf{l}^i \in \mathbb{I}_i^{rs}} d(\boldsymbol{\mu}_i^u, \mathbf{l}^i) \tag{5}$$

If $d^{rs}$ in the current iteration decreases compared to the previous iteration, we update the unlabeled data prototypes; otherwise, they remain unchanged. This refinement strategy also enhances the quality of pseudo labels generated for unlabeled data using $\hat{y}_i^u = \arg\min_k d(\boldsymbol{\mu}_k^u, \phi(\mathbf{x}_i^u))$.

### 3.3 PROTOTYPICAL CONTRASTIVE LEARNING

In prototypical contrastive learning (PCL), we assume that labeled instances from common categories share highly similar characteristics with unlabeled data prototypes representing the same common categories ($\boldsymbol{\mu}_{com}^u$), the same for unlabeled instances and labeled data prototypes. In this regard, we select out labeled and unlabeled instances from common categories using the decouple prototypes ($\boldsymbol{\mu}_{pk}$) and pseudo-labels ($\hat{y}_i^u$), respectively. These instances subsequently yield feature sets $\mathbb{F}_l = \{\mathbf{f}^l, \mathbf{f}_{com}^u\}$ and $\mathbb{F}_u = \{\mathbf{f}_{com}^l, \mathbf{f}^u\}$, leading to semi-supervised PCL setting (Figure 3 (a)).

For each feature $\mathbf{f}_i^u \in \mathbb{F}_u$, we calculate its prototypical distribution with respect to unlabeled data prototypes, represented as $P_i^u = \begin{bmatrix} p_{i,1}^u & \cdots & p_{i,||\mathbb{C}_u||_1}^u \end{bmatrix}$, following Eq. 6a. We then compute the prototypical contrastive loss for unlabeled data prototypes using their distributions and cluster label $c_{i,j}^u$ (Eq. 6b). A similar loss is applied to $\mathbb{F}_l$, resulting in the overall PCL loss (Eq. 6c).

$$p_{i,j}^u = \frac{\exp(\boldsymbol{\mu}_j^u \cdot \mathbf{f}_i^u / \tau)}{\sum_{\boldsymbol{\mu}^u \in H_u} \exp(\boldsymbol{\mu}^u \cdot \mathbf{f}_i^u / \tau)} \tag{6a}$$

$$\mathcal{L}_{\text{PCL}_u}(\mathbb{F}_u) = -\frac{1}{||\mathbb{F}_u||_1} \sum_{i=1}^{||\mathbb{F}_u||_1} \sum_{j=1}^{||\mathbb{C}_u||_1} c_{i,j}^u \log p_{i,j}^u \tag{6b}$$

$$\mathcal{L}_{\text{PCL}} = \lambda_l \mathcal{L}_{\text{PCL}_l}(\mathbb{F}_l) + \mathcal{L}_{\text{PCL}_u}(\mathbb{F}_u) \tag{6c}$$

where $\lambda_l$ controls the contribution of PCL loss $w.r.t$ labeled data prototypes.

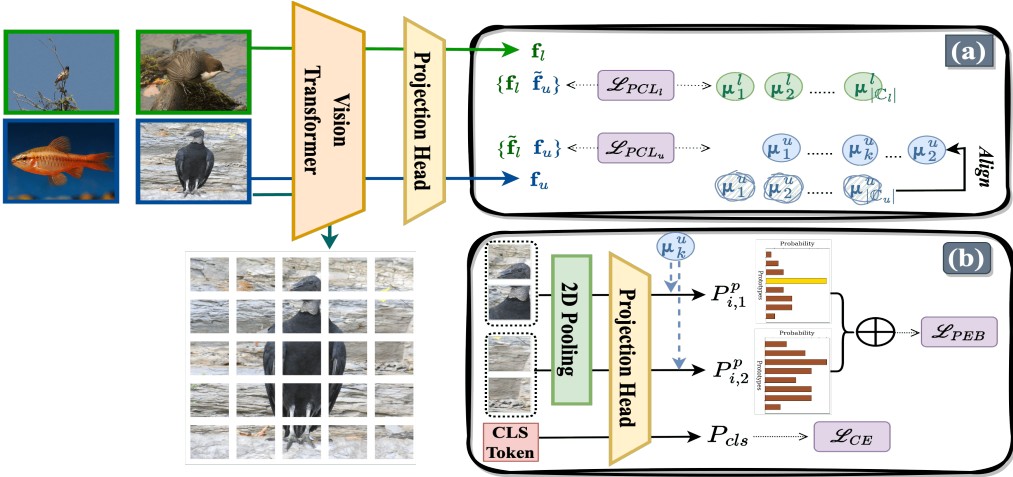

Figure 3: **Overview of our prototypical network**. (a) Prototypical Contrastive Learning (PCL) with Prototype Refinement (PR) Strategy. (b) Patch-Entropy Balance (PEB) method.

## 3.4 PATCH-ENTROPY BALANCE FOR SEMANTIC ENRICHMENT

As mentioned before, the classification of complex images requires enriching semantic guidance during the optimization of PCL. Also, a complex image can be considered as a composition of various levels of descriptions: a concept word (`cls token`) and several detail-oriented words (`patch tokens`), where each patch token conveys specific local information about the image. Building upon this concept, we represent each image using multiple patch tokens and attach a prototypical distribution to each patch. This approach results in the creation of patch-level prototypical distributions, which guide prototypes to pay attention to diverse characteristics, thereby enhancing semantics awareness during optimization.

However, there's a trade-off: smaller patches provide local details, enhancing the model's generalization capability during pretraining but increasing computational cost, and larger patches contain more global semantic information, offering better guidance in specific classification tasks. To balance this, we use 2D pooling on patch-tokens to fuse adjacent representations (Figure 3 (b)), then multiplied the outputs with prototypes to generate the patch-level prototypical distributions $P_{i,j}^p$ (prototypical distributions of $j$-th patch in image $\mathbf{x}_i$). The trade-off is further analyzed in Appendix G.

We notice that some patches have sparse prototypical distributions, containing category-relevant information, while others have denser distributions, offering auxiliary or even irrelevant disturbances. To enhance category-related semantics and reduce irrelevant noise, we introduce the Patch-Entropy Balance (PEB) loss. PEB loss achieves a balance between sparsity and uniformity across all patch-level prototypical distributions. It combines self-entropy loss for sparsity promotion and Kullback–Leibler divergence from uniformity (KLDU) loss for uniformity enhancement. Specifically, PEB dynamically encourages sparse distributions to become even sparser and dense distributions to approach uniformity. The PEB loss is formulated as follows:

$$\mathcal{L}_{\text{PEB}} = \frac{1}{||\mathbb{D}_{tr}||_1 \times N_p} \sum_i^{||\mathbb{D}_{tr}||_1} \sum_j^{N_p} \left[ H(P_{i,j}^p) + \lambda_{\text{uni}}(\alpha_i^j P_{i,j}^p \log(\frac{P_{i,j}^p}{U_c})) \right] \quad (7)$$

$$H(P_{i,j}^p) = - P_{i,j}^p \log P_{i,j}^p \quad \text{and} \quad \alpha_i^j = \exp(H(P_{i,j}^p))$$

In this formula, $H(P_{i,j}^p)$ quantifies distribution entropy, and $\lambda_{\text{uni}}$ is the weight for KLDU loss. $\alpha_i^j$ modulates the KL divergence term based on the initial distribution's entropy in patch $\mathbf{x}_i^j$, with larger values for higher entropy (uniform distributions) and smaller values for lower entropy (sparse distributions). The PEB method is further analyzed theoretically in Appendix D.

As mentioned before, the patch-level tokens capture intricate image details, while classification tokens (`cls-token`) focus on category concepts. To further boost model performance, we incorporate

cross-entropy loss for both labeled and unlabeled instances as follows:

$$\mathcal{L}_{\text{CE}} = -\frac{1}{||\mathbb{D}_{tr}||_1} \sum_{i}^{||\mathbb{D}_{tr}||_1} P_{GT}(\mathbf{x}_i^j) \log(P_{cls}(\mathbf{x}_i)) \qquad (8)$$

Here, $P_{cls}$ represents predicted probabilities of the `cls-token`, and $P_{GT}$ stands for ground-truth category probabilities. For unlabeled instances $\mathbf{x}^u$, we use pseudo labels $\hat{y}_i^u$ as ground truth.

In consequence, the overall loss of our prototypical framework is formulated as follows:

$$\mathcal{L}_{\text{SEPNet}} = \lambda_{\text{PCL}} \mathcal{L}_{\text{PCL}} + \lambda_{\text{PEB}} \mathcal{L}_{\text{PEB}} + \lambda_{\text{CE}} \mathcal{L}_{\text{CE}} \qquad (9)$$

## 4 Experiments

### 4.1 Data and Experimental Details

With the popular datasets in NCD task: Cifar10, Cifar100 (Krizhevsky et al., 2009) and ImageNet100 (Deng et al., 2009), we split the categories of each dataset into private-known, common, and novel categories ($\mathbb{C}_{pk}, \mathbb{C}_{com}, \mathbb{C}_{pk}$) to construct the UniNCD task. The splitting ratio are $|\mathbb{C}_{com}|/|\mathbb{C}_{pk}|/|C_n| = 6/2/2$ for Cifar10, $60/20/20$ for Cifar100 and ImageNet100. We then select 50% of common categories with all private-known categories to form the labeled dataset and keep the remaining to form the unlabeled dataset. Implementation details are discussed in Appendix H.

**Comparison of Methods.** We compared our method with ten methods: (1) Deep Transfer Clustering (DTC) (Han et al., 2019), (2) Ranking Statistics (RS) (Han et al., 2020), (3) Divide and Conquer (CompEX) (Yang et al., 2022), (4) NCD Spectral Contrastive Loss (NSCL) (Sun et al., 2023), (5) Inter-class and Intra-class Constraints (IIC) (Li et al., 2023), which are methods mainly for disjoint NCD; The methods for Open-World NCD are (6) Open-World Semi-Supervised Learning (ORCA) (Cao et al., 2021), (7) Generalized Category Discovery (GCD) (Vaze et al., 2022), (8) Open-World Contrastive Learning (OpenCon) (Sun & Li, 2022), (9) Contrastive Affinity Learning (PromptCAL) (Zhang et al., 2023), and (10) Decoupled Prototypical Networks (DPN) (An et al., 2023).

**Evaluation Protocols.** The performance was evaluated by measuring accuracy between the model's cluster assignments and ground-truth labels on the test set, with three aspects: all instances (All), instances from known categories (Known), and instances from novel categories (Novel). The number of categories in the unlabeled dataset ($\mathbb{C}_u$) is often unknown. Following previous studies (Xie et al., 2016; Han et al., 2020; Sun & Li, 2022), we set $K$ (cluster number) equal to $\mathbb{C}_u$ based on previous studies, as approximate cluster estimation is usually feasible in the real world. We also discuss estimating the number of categories in unlabeled datasets in Appendix F.2.

### 4.2 Experiments Results and Discussion

#### 4.2.1 Main Results.

Our method consistently outperforms other NCD methods across all datasets on the test set, with an average improvement of 2.96% (Known), 6.29% (Novel), and 3.34% (All) over the top-performing baseline (Table 1). This reflects our model's efficacy in handling both known and novel categories. Additionally, our strategy excels in novel category accuracy, demonstrating its ability to mitigate negative transfer risk. This can be mainly attributed to the fact that our prototypical framework considers and excludes misaligned prototypes. By doing so, we manage to obtain higher-quality supervision signals for prototypical contrastive learning. We also try to improve DPN (An et al., 2023) with Prototype Refinement strategy ("DPN w/ PR"), which indeed brings improvement. However, its performance still lags behind ours by a large margin. Furthermore, on the complex dataset, Cifar100, which mirrors the unpredictability of real-world conditions, our method distinctly outperformed previous strategies by a larger margin than the other two datasets.

### 4.3 Ablation Study

We have three Prototype Refinement strategy variations: "w/o PR" (prototype alignment without refinement), "w/o filtering" (refining prototypes without Z-score filtering), and "w/o update" (no

Table 1: **Evaluation results (%) for all, known and novel categories**. Asterisk (∗) denotes that the original method can not recognize known categories. Results on ORCA, GCD, OpenCon, PromptCAL, DPN, and SEPNet (Ours) are averaged over three runs with different random seeds.

| Method | Cifar10 | | | Cifar100 | | | ImageNet100 | | |
|---|---|---|---|---|---|---|---|---|---|
| | Known | Novel | All | Known | Novel | All | Known | Novel | All |
| ∗DTC | 50.63 | 39.54 | 48.41 | 29.65 | 12.52 | 26.22 | 22.89 | 15.54 | 21.42 |
| ∗RS | 70.56 | 52.29 | 66.91 | 26.35 | 18.68 | 24.81 | 45.00 | 24.55 | 40.91 |
| ∗CompEX | 76.72 | 62.20 | 73.82 | 33.01 | 24.83 | 31.37 | 61.03 | 41.86 | 57.20 |
| ∗NSCL | 79.26 | 69.53 | 77.31 | 39.82 | 30.20 | 37.90 | 66.15 | 47.84 | 62.49 |
| ∗IIC | 82.10 | 72.27 | 80.13 | 44.16 | 34.60 | 42.25 | 71.81 | 52.68 | 68.00 |
| ORCA | $85.24^{\pm0.1}$ | $79.41^{\pm0.4}$ | $84.07^{\pm0.2}$ | $63.95^{\pm0.3}$ | $39.59^{\pm1.0}$ | $59.07^{\pm0.8}$ | 85.46 | 66.34 | 81.64 |
| GCD | $86.12^{\pm0.4}$ | $79.04^{\pm0.5}$ | $84.70^{\pm0.5}$ | $66.25^{\pm0.5}$ | $40.61^{\pm0.9}$ | $61.12^{\pm0.6}$ | 86.79 | 69.79 | 83.39 |
| OpenCon | $86.78^{\pm0.2}$ | $84.11^{\pm0.4}$ | $86.25^{\pm0.2}$ | $66.13^{\pm0.4}$ | $43.51^{\pm0.6}$ | $61.06^{\pm0.4}$ | 86.56 | 74.13 | 84.07 |
| PromptCAL | $87.29^{\pm0.7}$ | $85.55^{\pm1.0}$ | $86.94^{\pm0.8}$ | $67.32^{\pm1.0}$ | $45.24^{\pm1.4}$ | $62.94^{\pm1.2}$ | 86.96 | 75.33 | 84.63 |
| DPN | $85.74^{\pm0.4}$ | $78.52^{\pm0.8}$ | $84.30^{\pm0.5}$ | $64.20^{\pm0.7}$ | $40.55^{\pm0.9}$ | $59.47^{\pm0.7}$ | 86.32 | 68.23 | 82.70 |
| DPN w/ PR | $87.35^{\pm0.7}$ | $83.25^{\pm1.0}$ | $86.53^{\pm0.8}$ | $65.58^{\pm1.1}$ | $44.92^{\pm1.5}$ | $61.45^{\pm1.2}$ | 86.66 | 73.00 | 83.93 |
| SEPNet | $\mathbf{91.47^{\pm1.2}}$ | $\mathbf{90.04^{\pm1.5}}$ | $\mathbf{91.18^{\pm1.3}}$ | $\mathbf{71.57^{\pm1.3}}$ | $\mathbf{48.97^{\pm1.6}}$ | $\mathbf{67.05^{\pm1.4}}$ | **87.40** | **82.98** | **86.51** |
| Improvement | +4.18 | +4.59 | +4.24 | +4.25 | +3.73 | +4.11 | +0.44 | +7.65 | +1.88 |

Table 2: **Accuracy for ablation study using Cifar10 dataset**. Bold and underlined indicate the best and worst performance, respectively.

| Method | Known | Novel | All |
|---|---|---|---|
| Ours | **91.47** | **90.04** | **91.18** |
| w/o PR | 86.59 | 77.90 | 84.84 |
| w/o filtering | 87.24 | 80.13 | 85.02 |
| w/o update | 91.05 | 84.52 | 89.72 |
| w/o $\mathcal{L}_{PEB}$ | 86.48 | 80.60 | 85.30 |
| w/o PPD | 87.02 | 82.86 | 86.19 |
| w/o KLDU | 90.20 | 87.32 | 89.63 |

Table 3: **Negative transfer risk (NTR) assessed on Cifar10 dataset**. Lower indicates mitigating negative transfer better.

| Method | Cifar10 | Cifar100 | ImageNet100 |
|---|---|---|---|
| ∗CompEX | 14.32 | 36.34 | 23.13 |
| ∗IIC | 11.24 | 28.09 | 19.36 |
| GCD | 8.08 | 19.72 | 11.90 |
| OpenCon | 9.09 | 20.30 | 13.91 |
| PromptACL | 6.61 | 14.89 | 9.37 |
| DPN | 10.16 | 22.38 | 14.13 |
| SEPNet | **2.41** | **12.53** | **5.38** |

iterative updates of unlabeled data prototypes). Additionally, there are three Patch-Entropy Balance method versions: "w/o $\mathcal{L}_{PEB}$" (loss computation without patch-entropy balance loss), "w/o PPD" (computing $\mathcal{L}PEB$ using instance-level prototypical distributions), and "w/o KLDU" (using self-entropy loss without KL divergence from uniformity loss in $\mathcal{L}_{PEB}$). Table 2 shows that $\mathcal{L}_{PEB}$ significantly impacts known categories, while PR strategy primarily affects novel and all categories.

## 4.4 EXPERIMENTAL ANALYSIS

**Negative Transfer Risk (NTR).** Negative transfer in UniNCD primarily arises when instances belonging to novel categories are misclassified as private-known categories. Following Wang et al. (2019), with the dataset consisting of all instances from novel categories, denoted as $\mathbb{D}_n$, and the model $\phi$, we define a measurable metric to access the NTR in the UniNCD task as follows:

$$R_{NT}(\phi) := \mathbb{E}_{\mathbf{x} \sim \mathbb{D}_n}[\varphi(\phi(\mathbf{x})) \in \mathbb{C}_{pk}] \tag{10}$$

where $\varphi$ represents $K$-Means algorithm and $\mathbb{C}_{pk}$ represents the set of private-known categories. Table 3 shows NTR comparisons between our method and six other NCD methods across three datasets. Our method consistently achieves significantly lower NTR on all datasets, highlighting its effectiveness in addressing the UniNCD task. Notably, OpenCon (Sun & Li, 2022) and DPN (An et al., 2023) exhibit higher NTR than other Open-World NCD methods, potentially due to their explicit decoupling of unlabeled data into known and novel categories, which increases NTR. In contrast, our method decouples categories with a focus on private-known categories, avoiding optimizing representations guided by misaligned prototypes, resulting in lower NTR.

**Effectiveness of Prototype Refinement (PR) strategy.** To verify the efficacy of our PR strategy, we employ PCA and t-SNE to visualize the prototypes at epoch 1, before refinement, and at epoch

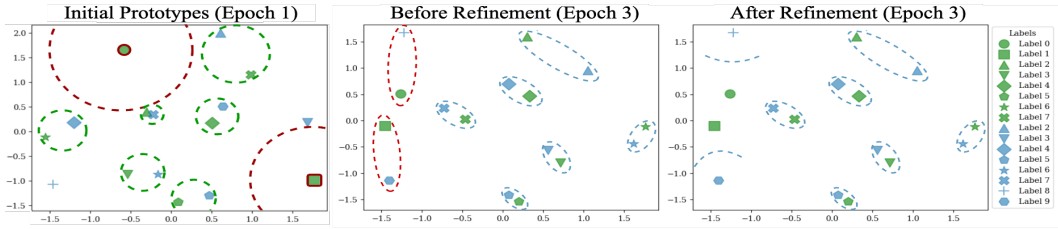

Figure 4: **t-SNE on prototypes of Cifar10**: impact of Prototype Refinement strategy.

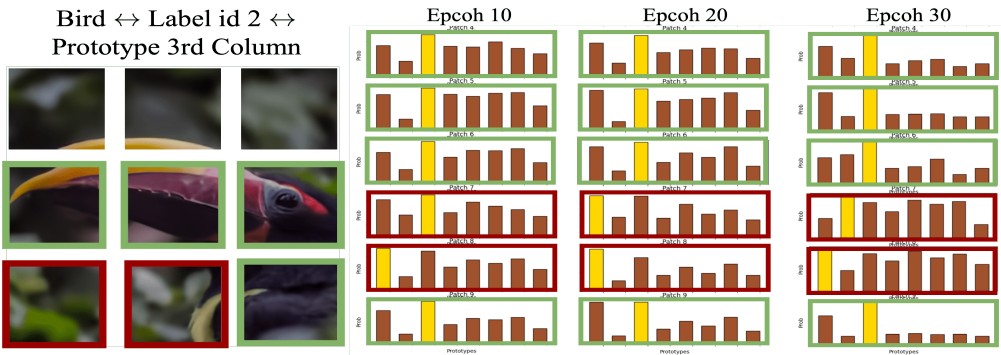

Figure 5: **Changes in patch-level prototypical distributions during training**. Left: the image from Cifar10 dataset is (enhanced by super-resolution for visualization) encoded into 3 x 3 patches. Right: prototypical distributions from patch 4 to 9 over epochs 10, 20, and 30.

3 (both before and after refinement). As depicted in Figure 4, initial private-known-category prototypes are distant from other prototypes (circled in scarlet), compared to common-category prototypes (circled in green). Furthermore, at the beginning of training, the representations lack discriminative qualities, leading to several misaligned prototypes. With three epochs of refinement, common-category prototypes are aligned correctly (circled in blue), but novel-category prototypes remain misaligned with private-known-category prototypes (circled in red), resulting in negative transfer (Figure 4, middle). With PR strategy, these misaligned prototypes are successfully excluded from the matching pairs, achieving optimal prototype alignment (Figure 4, right). The stability of PR strategy across different datasets is discussed in Appendix F.1.

**Sparsity of Patch-level Prototypical Distributions.** To further validate its effectiveness, we visualize the changes in patch-level prototypical distributions at epochs 10, 20, and 30 using an image from Cifar10 (enhanced using super-resolution for clearer visualization). We focus on patches 4, 5, 6, and 9, which contain category-related information (highlighted in green), and patches 7 and 8, which contain irrelevant information. We observe that the prototypical distributions of these informative patches become sparser (highlighted in green). The prototype corresponding to their respective categories exhibits a higher peakiness, indicating their increased contribution to prototype learning. Notably, the prototype distribution of patch 7 is initially concentrated on category-related prototypes and gradually becomes less category-related after several epochs of training.

## 5 CONCLUSIONS

We present a *Semantic-Enhanced Prototypical Network* (SEPNet) for Universal Novel Category Discovery (UniNCD). SEPNet addresses the challenges of prototype misalignment faced by previous prototype-based methods. With the *Prototype Refinement* (PR) strategy and the *Patch-Entropy Balance* (PEB) method, our framework enhances the distinctiveness of instances to prototypes while avoiding negative transfer. Experimental results across three benchmark datasets demonstrated the effectiveness of SEPNet, surpassing state-of-the-art methods by a large margin.

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

# A    PRETRAINING LOSS

## A.1    GENERALIZED CONTRASTIVE LOSS

Following the recent work (Sun & Li, 2022), we first introduce a generalized contrastive loss that can provide the foundation for the loss in the pretraining stage. The contrastive loss given anchor point $\mathbf{x}$ can be formulated as:

$$\mathcal{L}_\phi(\mathbf{x}) = -\frac{1}{||\mathbb{P}(\mathbf{x})||_1} \sum_{\mathbf{f}^+ \in \mathbb{P}(\mathbf{x})} \log \frac{\exp(\mathbf{f}^\top \cdot \mathbf{f}^+ / \tau)}{\sum_{\mathbf{f}^- \in \mathbb{N}(\mathbf{x})} \exp(\mathbf{f}^\top \cdot \mathbf{f}^- / \tau)} \tag{11}$$

where, $\tau$ is the temperature parameter, $\mathbb{P}(\mathbf{x})$ and $\mathbb{N}(\mathbf{x})$ are the positive and negative set of feature embeddings, respectively.

## A.2    SUPERVISED CONTRASTIVE LOSS (SCL)

For a mini-batch $\mathbb{B}_l$ with instances drawn from $\mathbb{D}_l$, two random augmentations are applied for each instance and construct a two-viewed batch $\tilde{\mathbb{B}}_l$, with its feature embeddings batch denoted as $\mathbb{A}_l$. Then for any instance $\mathbf{x}$ in $\mathbb{B}_l$, the positive and negative set can be expressed as:

$$\begin{cases} \mathbb{P}_l(\mathbf{x}_a) = \{\mathbf{f}' \mid \mathbf{f}' \in \{\mathbb{A}_l \backslash \mathbf{f}_a\}, y' = y_a\} \\ \mathbb{N}_l(\mathbf{x}_a) = \mathbb{A}_l \backslash \mathbf{f}_a \end{cases} \tag{12}$$

Subsequently, the supervised contrastive loss can be defined as:

$$\mathcal{L}_{\text{SCL}}(\mathbb{D}_l) = \sum_{\mathbf{x} \in \tilde{\mathbb{B}}_l} \mathcal{L}_\phi(\mathbf{x}; \tau_l, \mathbb{P}_l(\mathbf{x}), \mathbb{N}_l(\mathbf{x}))) \tag{13}$$

where $\tau_l$ is the temperature.

## A.3    SELF-SUPERVISED CONTRASTIVE LOSS (SSCL)

Similarly, for any instance $\mathbf{x}$ taken as the anchor point $\mathbf{x}_a$ in $\mathbb{B}_u$, the positive and negative set can be expressed as:

$$\begin{cases} \mathbb{P}_u(\mathbf{x}_a) = \{\mathbf{f}' \mid \mathbf{f}' = \psi(\mathbf{x}_a)\} \\ \mathbb{N}_u(\mathbf{x}_a) = \mathbb{A}_u \backslash \mathbf{f}_a \end{cases} \tag{14}$$

where $\psi : \mathcal{X} \to \mathcal{X}$ is an augmentation function.

The self-supervised contrastive loss is then defined as:

$$\mathcal{L}_{\text{SSCL}}(\mathbb{D}_u) = \sum_{\mathbf{x} \in \tilde{\mathbb{B}}_u} \mathcal{L}_\phi(\mathbf{x}; \tau_u, \mathbb{P}_u(\mathbf{x}), \mathbb{N}_u(\mathbf{x}))) \tag{15}$$

where $\tau_u$ is the temperature.

# B    OVERALL PROCESS

We present a general outline of our method, SEPNet, in Algorithm 1. Furthermore, to offer a visual representation of the framework's structure, we provide an illustrative overview in Fig. 6.

# C    THEORETICAL ANALYSIS OF OUR PROTOTYPICAL NETWORK

Our UniNCD learning objective can be effectively understood through the lens of the Expectation-Maximization (EM) algorithm. This perspective enables us to break down our method into two main steps: the E-step and the M-step. In the E-step, exemplified here with unlabeled data, we assign each instance $\mathbf{x}^u \in \mathbb{D}_u$ to a specific prototype based on the refined unlabeled data prototypes $\mu^u$. Following this, in the M-step, the EM algorithm aims to maximize the likelihood under the posterior class probabilities obtained from the E-step.

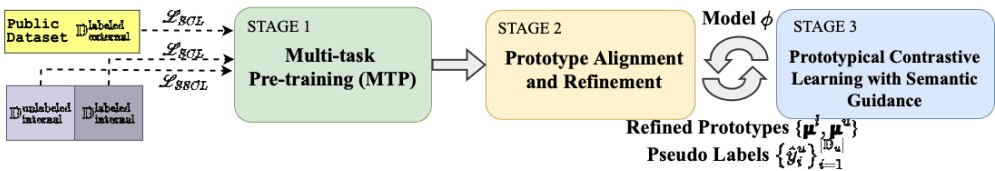

Figure 6: **Overall process of our proposed method**.

---

**Algorithm 1** Semantic-Enhanced Prototypical Network

---

**Input**: Labeled public sets $\mathbb{D}_{ex}^l$, labeled internal set $\mathbb{D}_l = \{\mathbf{x}_i^l, y_i^l\}_{i=1}^{N_l}$ and unlabeled internal set $\mathbb{D}_u = \{\mathbf{x}_i^u\}_{i=1}^{N_u}$, neural network encoder $\phi$

**Training Stage**:

  1: Pretrain $\phi$ with $\mathcal{L}_{\text{SCL}}$ on $\mathbb{D}_{ex}^l \cup \mathbb{D}_l$, and $\mathcal{L}_{\text{SSCL}}$ on $\mathbb{D}_u$.
  2: **while** convergence **do**
  3:    Update feature and label memory via $\phi$
  4:    **if** epoch = updating epoch **then**
  5:       Update labeled data prototypes $\{\boldsymbol{\mu}_j^l\}_{j=1}^{||\mathbb{C}_l||_1}$
  6:    **end if**
  7:    **if** first $N$ epochs **then**
  8:       **Prototype Alignment and Refinement:**
  9:       Align prototypes $\{\boldsymbol{\mu}_j^l\}_{j=1}^{||\mathbb{C}_l||_1}$ and $\{\boldsymbol{\mu}_j^u\}_{j=1}^{||\mathbb{C}_u||_1}$ using bipartite matching
10:       Rectify unlabeled data prototypes $\{\boldsymbol{\mu}_j^u\}_{j=1}^{||\mathbb{C}_u||_1}$ using Prototype Refinement strategy
11:       Assign pseudo labels $\hat{y}_i^u$ by prototypes to $\mathbf{x}_i^u$
12:    **end if**
13:    **Enriched Semantic Guidance:**
14:    Encode images into patch tokens and class tokens
15:    Calculate patch-entropy balance loss $\mathcal{L}_{\text{PEB}}$ using patch tokens
16:    Calculate prototypical contrastive loss $\mathcal{L}_{\text{PCL}}$ and cross-entropy loss $\mathcal{L}_{\text{CE}}$ using class tokens
17:    Employ overall loss $\mathcal{L}_{\text{SEPNet}}$ to update $\phi$ using back-propagation
18: **end while**
19: **return** $\phi$

---

The theoretical foundation of our approach suggests that minimizing our loss (Eq. 9) partially maximizes the likelihood by clustering similar instances. Specifically, our loss encourages the concentration of similar instances around the corresponding prototypes, promoting more compact representations.

## C.1 E-STEP (PROTOTYPE ALIGNMENT AND REFINEMENT)

In the E-step, the EM algorithm's objective is to maximize the likelihood using the encoder $\phi$ and prototype matrix $\boldsymbol{H}_u$, which can be lower bounded as follows:

$$\sum_i^{||\mathbb{D}_u||_1} \log p(\mathbf{x}_i^u \mid \phi, \boldsymbol{H}_u) \geq \sum_i^{||\mathbb{D}_u||_1} q_i(c) \log \sum_{c \in C_u} \frac{p(\mathbf{x}_i^u, c \mid \phi, \boldsymbol{H}_u)}{q_i(c)} \tag{16}$$

Here, $q_i(c)$ represents the prototypical distribution for instance $\mathbf{x}_i^u$ *w.r.t.* category $c$, reflecting the posterior class probability. Based on the concavity of $\log(\cdot)$ and $p(\mathbf{x}_i^u \mid \phi, \boldsymbol{H}_u)$ in Eq. 16, we can estimate $q_i(c)$ as Eq. 17.

$$q_i(c) = \frac{p(\mathbf{x}_i^u, c \mid \phi, \boldsymbol{H}_u)}{\sum_{c \in \mathbb{C}_u} p(\mathbf{x}_i^u, c \mid \phi, \boldsymbol{H}_u)} = \frac{p(\mathbf{x}_i^u, c \mid \phi, \boldsymbol{H}_u)}{p(\mathbf{x}_i^u \mid \phi, \boldsymbol{H}_u)} = p(c \mid \mathbf{x}_i^u, \phi, \boldsymbol{H}_u) \tag{17}$$

To estimate $p(c \mid \mathbf{x}_i^u, \phi, \boldsymbol{H}_u)$, we follow Fisher (1953) to model the data with *von Mises-Fisher* (vMF) distribution, which is well-suited for our high-dimensional hyperspherical space considering

$\phi(\mathbf{x}^u)$. This allows us to express $p(c \mid \mathbf{x}_i^u, \phi, \boldsymbol{H}_u)$ as $\text{Softmax}(\boldsymbol{H}_u^\top \cdot \phi(\mathbf{x}_i^u) \mid c)$. Additionally, with the help of pseudo labels $\hat{y}_i$, we can further simplify $q_i(c)$ as $q_i(c) = \mathbf{1}_{\text{condition}}\{\hat{y}^i = c\}$.

## C.2 M-step (Prototypical Contrastive Learning with Enriched Semantic Guidance)

During the M-step, we optimize our prototypical network $\phi$ and the prototype matrix $\boldsymbol{H}_u$ based on the prototypical distribution $q_i(c)$ derived from the E-step. And by defining $\tilde{\mathbb{D}}_u(c) = \{\mathbf{x}_i^u \in \mathbb{D}_u \mid \hat{y}_i = c\}$, we can convert the maximization into a joint optimization objective as follows:

$$\arg \max_{\phi, \boldsymbol{H}_u} \sum_{i=1}^{||\mathbb{D}_u||_1} \sum_{c \in \mathbb{C}_u} q_i(c) \log \frac{p(\mathbf{x}_i^u, c \mid \phi, \boldsymbol{H}_u)}{q_i(c)}$$

$$= \arg \max_{\phi, \boldsymbol{H}_u} \sum_{i=1}^{||\mathbb{D}_u||_1} \sum_{c \in \mathbb{C}_u} q_i(c) \log p(\mathbf{x}_i^u \mid c, \phi, \boldsymbol{H}_u) \qquad \text{Because of } q_i(c) \log \frac{p(c)}{q_i(c)} \text{ is a constant}$$

$$= \arg \max_{\phi, \boldsymbol{H}_u} \sum_{i=1}^{||\mathbb{D}_u||_1} \sum_{c \in \mathbb{C}_u} \mathbf{1}_{\text{condition}}\{\hat{y}_i = c\} \log p(\mathbf{x}_i^u \mid c, \phi, \boldsymbol{H}_u)$$

$$= \arg \max_{\phi, \boldsymbol{H}_u} \sum_{c \in \mathbb{C}_u} \sum_{\mathbf{x}^u \in \tilde{\mathbb{D}}_u(c)} \log p(\mathbf{x}^u \mid c, \phi, \boldsymbol{H}_u) \qquad \text{Because of indexes } c \text{ can be merged with } \mathbb{D}_u$$

$$= \arg \max_{\phi, \boldsymbol{H}_u} \sum_{c \in \mathbb{C}_u} \sum_{\mathbf{x}^u \in \tilde{\mathbb{D}}_u(c)} \phi(\mathbf{x}^u)^\top \cdot \boldsymbol{\mu}_c^u \qquad \text{Because of vMF density function}$$

The final expression above can be interpreted as aligning the representation $\phi(\mathbf{x})$ with the corresponding prototype $\boldsymbol{\mu}_c^u$. Notably, our algorithm achieves this maximization step by separately optimizing $\boldsymbol{H}_u$ and $\phi$ as follows:

1. Optimizing $\boldsymbol{H}_u$:
   By fixing $\phi$, we can obtain the optimal prototype $\hat{\boldsymbol{\mu}}^u$ through our Prototype Refinement strategy.

2. Optimizing $\phi$:
   By fixing $\boldsymbol{H}_u$, we aim to demonstrate minimize $\mathcal{L}_{\text{PCL}_u}$ resembles the joint optimization objective in M-step. To achieve this, we decompose $\mathcal{L}_{\text{PCL}_u}$ into two parts as follows:

$$\mathcal{L}_{\text{PCL}_u}(\mathbb{F}_u) = -\frac{1}{||\mathbb{F}_u||_1} \sum_{i=1}^{||\mathbb{F}_u||_1} \sum_{j=1}^{||\mathbb{C}_u||_1} c_{i,j}^u \log \frac{\exp(\boldsymbol{\mu}_j^u \cdot \mathbf{f}_i^u / \tau)}{\sum_{\mu^u \in \boldsymbol{H}_u} \exp(\mu^u \cdot \mathbf{f}_i^u / \tau)}$$

$$= \underbrace{-\frac{1}{||\mathbb{F}_u||_1} \sum_{i=1}^{||\mathbb{F}_u||_1} \sum_{j=1}^{||\mathbb{C}_u||_1} c_{i,j}^u (\boldsymbol{\mu}_j^u \cdot \mathbf{f}_i^u / \tau)}_{\mathcal{L}_{\text{align}}(\mathbf{x})} + \frac{1}{||\mathbb{F}_u||_1} \sum_{i=1}^{||\mathbb{F}_u||_1} \sum_{j=1}^{||\mathbb{C}_u||_1} c_{i,j}^u \sum_{\mu \in \boldsymbol{H}_u} \exp(\mu^u \cdot \mathbf{f}_i^u / \tau)$$

$$\tag{18}$$

Notably, minimizing $\mathcal{L}_{\text{align}}$ (Eq. 18) in the PCL loss $\mathcal{L}_{\text{PCL}}$ promotes the proximity of representations to their most similar prototypes. This effectively approximates the maximization of the objective outlined with optimal prototypes $\hat{\boldsymbol{\mu}}^u$ as follows:

$$\arg \min_\phi \sum_{\mathbf{x}^u \in \mathbb{D}_u} \mathcal{L}_{\text{align}}(\mathbf{x}^u) = \arg \max_\phi \sum_{c \in \mathbb{C}_u} \sum_{\mathbf{x}^u \in \mathcal{S}(x)} \phi(\mathbf{x}^u)^\top \cdot \hat{\boldsymbol{\mu}}^u \tag{19}$$

These observations validate that our prototypical network learns representations for novel classes in an EM fashion. Importantly, we extend EM from a traditional learning setting to the UniNCD task with the capability to handle universal world data.

## D Theoretical Analysis of Patch-Entropy Balance Method

The Patch-Entropy Balance (PEB) method aims to enhance category-related semantics in sparse distributions and reduce irrelevant noise in dense distributions. It achieves a balance between

sparsity and uniformity by combining self-entropy loss for sparsity promotion and Kullback-Leibler Divergence from Uniformity (KLDU) loss for uniformity enhancement.

### D.1 Self-Entropy Loss (Sparse Distribution Enhancement)

The term $H(P_{i,j}^p) = -P_{i,j}^p \log P_{i,j}^p$ calculates the self-entropy of the patch-level prototypical distribution $P_{i,j}^p$. This term encourages sparsity by increasing peakiness in the distribution. When $P_{i,j}^p$ is sparse (i.e., one or a few categories have significantly higher probabilities), the entropy is lower, and the self-entropy loss penalizes low probabilities, pushing the distribution towards sparsity.

### D.2 Kullback-Leibler Divergence from Uniformity (KLDU) Loss (Uniformity Enhancement)

The term $\alpha_i^j P_{i,j}^p \log \left( \frac{P_{i,j}^p}{U_c} \right)$ represents the KLDU loss, which measures how far $P_{i,j}^p$ is from a uniform distribution $U_c$ (where $U_c$ represents uniformity).

- Weighting Function $\alpha_i^j$: The term $\alpha_i^j$ is a weighting function based on the initial distribution's entropy in patch $\mathbf{x}_i^j$. It is computed as $\alpha_i^j = \exp(H(P_{i,j}^p))$. The underlying concept is that $\alpha_i^j$ tends to be larger for uniform distributions (high entropy), making the KL divergence term have a larger impact. Conversely, for sparse distributions with lower entropy, $\alpha_i^j$ tends to be smaller, effectively diminishing the impact of the KL divergence term.

- Effect on Sparse Distributions: For sparse distributions with low initial entropy, $\alpha_i^j$ is small, reducing the impact of the KLDU loss. This means that the loss has a minimal effect on distributions that are already sparse. It avoids overly penalizing them and preserves their peakiness.

- Effect on Dense Distributions: For dense distributions that are already close to uniform (high initial entropy), $\alpha_i^j$ becomes larger, making the KLDU loss have a more substantial impact. It encourages adjustments to bring these distributions closer to uniformity, effectively reducing peakiness and making them more uniform.

### D.3 Overall PEB Loss

The PEB loss combines both the self-entropy loss and the KLDU loss. By doing so, it seeks to strike a balance between sparsity promotion and uniformity enhancement.

- Balancing Sparsity and Uniformity: The self-entropy loss encourages peakiness reduction and sparsity promotion. The KLDU loss encourages uniformity enhancement. By combining these two components, the PEB loss ensures that the resulting distribution is not overly skewed towards specific categories (due to peakiness) while also being sufficiently deterministic to minimize uncertainty.

- Regularization and Noise Reduction: The KLDU loss acts as a regularization term that prevents overfitting to specific prototypes, which mitigates the catastrophic forgetting problem of categories, especially when encountering datasets with extensive and diverse categories. It discourages the model from assigning excessively high probabilities to irrelevant categories, effectively reducing noise and enhancing category-related semantics.

## E Effects of Known Category Ratio

In examining the ramifications of the known category ratio on the efficacy of our model, we manipulated this parameter in a range spanning from 0.3 to 0.7, with gradations of 0.1. Evidence gathered as displayed in Fig. 7, conclusively highlights our model's superior performance. It demonstrated supremacy across all categories, specifically novel categories. This strong showing underscores the validity and robustness of our model in UniNCD.

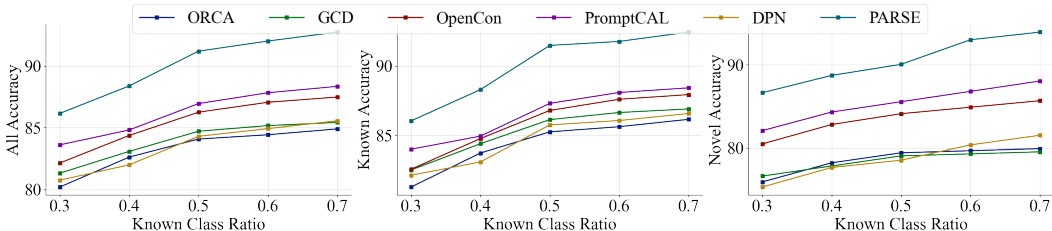

Figure 7: **Effects of know category ratios**. Clustering accuracies (%) for all, known, and novel categories are evaluated on the Cifar10 dataset.

Table 4: **Estimated number of private-known categories on Cifar100 and ImageNet100**. GT stands for ground truth, and the unit of error is the number of categories.

| Method | Cifar100 | ImageNet100 |
|--------|----------|-------------|
| GT | {10, 20, 40} | {10, 20, 40} |
| Ours | {10, 20, 38} | {10, 20, 39} |
| Error | 0.67 | 0.33 |

Table 5: **The results of estimating the number of categories on Cifar100 and ImageNet100**. Each number represents the number of categories in the unlabeled dataset.

| Method | Cifar100 | ImageNet100 |
|--------|----------|-------------|
| GT | {50, 60, 80} | {50, 60, 80} |
| DTC | {55, 68, 89} | {54, 65, 88} |
| Ours | {52, 59, 79} | {51, 59, 80} |
| Error | 1.33 | 0.67 |

## F  ESTIMATING THE NUMBER OF CATEGORIES

### F.1  NUMBER OF PRIVATE-KNOWN CATEGORIES

To evaluate the effectiveness of our Prototype Refinement strategy, we provide estimated values for private-known categories ($\mathbb{C}_{pk}$) on both the Cifar100 and ImageNet100 datasets, as displayed in Table 4. Our method displays impressive accuracy in estimating $\mathbb{C}_{pk}$, especially when it is relatively small. Even as $\mathbb{C}_{pk}$ approaches half of the total category count, our method exhibits only a minimal deviation (ranging from 2% to 5%) from the ground truth.

### F.2  NUMBER OF NOVEL CATEGORIES

We adopt the method outlined in DTC Han et al. (2019) to address the task of estimating the number of categories ($K$) from the unlabeled dataset $\mathbb{D}_u$. The outcomes are presented in Table 5, illustrating that our model achieves lower error rates on both Cifar100 and ImageNet100 compared to DTC. This highlights our model's competence in effectively utilizing information from both labeled and unlabeled data, leading to enhanced representations for accurate category number estimation.

## G  TRADE-OFF OF THE PATCH SIZE

As previously discussed, there's a trade-off between using smaller and larger patches in our approach. Smaller patches offer advantages during pre-training, enhancing generalization, but they also raise computational costs. We address this problem by applying 2D pooling to fuse adjacent representations. However, even with fusion, there remains a delicate balance to strike between UniNCD performance and computational efficiency. We systematically experimented with the pooling layer size, ranging from 2 to 8 patches, while conducting training on one 80GB A-100 GPU for 100 epochs.

As depicted in Figure 8, we observed that the time cost increases almost linearly concerning the number of patches, which is an expected outcome due to the heightened computational load. Conversely, the overall error decreases gradually with additional patches. This deceleration in error reduction suggests diminishing returns as more patches are added. Beyond a certain point, the extra patches may not substantially enhance discriminative information, resulting in a slower decrease in error.

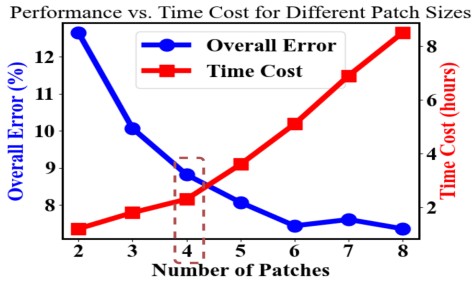

Figure 8: **Trade-off between UniNCD performance and computation cost with respect to the number of patches**. The left axis depicts the clustering error across all categories, while the right axis represents the total training time (in hours) over 100 epochs.

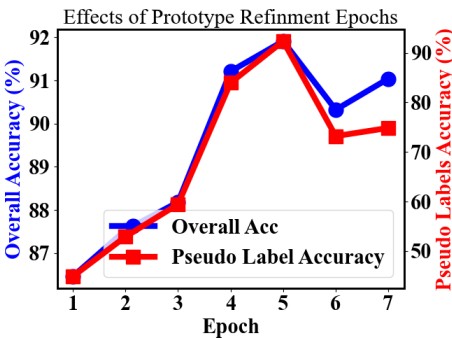

Figure 9: **Analysis of Prototype Refinement strategy epochs on Cifar10**. The left axis represents clustering accuracy across all categories, while the right axis shows the accuracy of pseudo labels.

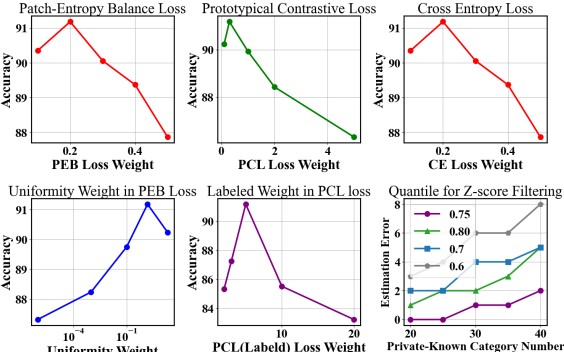

Figure 10: **Sensitivity analysis for weights of loss on Cifar10**. Subplots 1 ~ 5 (from left to right) depict five weights $\lambda_{PEB}$, $\lambda_{PCL}$, $\lambda_{CE}$, $\lambda_{uni}$ and $\lambda_l$ in the context of loss $\mathcal{L}_{SEPNet}$. Subplot 6 depicts the choice of the quantile $q$ for the Prototype Refinement strategy.

Considering real-world applications and the need for efficient incremental learning in novel category discovery, we opted for a pooling layer size of 4. This choice effectively balances the trade-off between computational cost and performance, making it a suitable compromise for practical use.

## H  SENSITIVITY ANALYSIS

We introduce the hyper-parameter settings for SEPNet and present the best combinations of hyper-parameters, including the PR strategy employing epochs and loss weights ($\mathcal{L}_{SEPNet}$), determined through a cross-validation strategy. The validation strategy involves splitting the classes in $\mathbb{D}_l$ equally into "known" and "novel" classes, with 50% of the instances from selected known classes labeled. We then use this new validation dataset to perform grid searching for selecting the best hyper-parameters, which are summarized in Table 6.

**Implementation Details.**  We employ pre-trained DINO (Caron et al., 2021) as our backbone network[1]. During pretraining, we use an early stopping strategy based on the performance of the validation set. For model optimization, we use AdamW optimizer, and the learning rate is set as 8e-3 for pretraining and 3e-4 for training. For the comparison methods, we split the datasets following our experimental setting and replace the backbones with the one we employed. Considering the trade-off, the pooling layer size in our method is set to 4, and other hyperparameters are selected using cross-validation, details will be discussed in the Appendix G. The basic training settings for Cifar10/Cifar100/ImageNet100 involve training the model for 80/100/150 epochs with batch sizes of 2048/2048/1536 using AdamW with coefficients $(0.9, 0.999)$ and a weight decay of 1e-2. The learning rate starts at 3e-4 and is updated using a cosine annealing schedule, while the momentum for feature updating ($m$) remains fixed at 0.95. We also employ a quantile of 0.75 for the Z-score filter.

---

[1]https://github.com/facebookresearch/dino

Table 6: **Hyperparameters groups selected using cross-validation**. "PR epochs" denotes the epochs using the Prototype Refinement strategy, $\lambda_{\text{PEB}}$, $\lambda_{\text{PCL}}$, $\lambda_{\text{CE}}$ and $\lambda_l$ are weights of loss in Eq. 9

| Dataset | PR epochs | $\lambda_{\text{PEB}}$ | $\lambda_{\text{PCL}}$ | $\lambda_{\text{CE}}$ | $\lambda_{\text{uni}}$ | $\lambda_l$ | $q$ |
|---|---|---|---|---|---|---|---|
| Cifar10 | 5 | 0.8 | 0.3 | 0.1 | 1.3 | 10.0 | 0.75 |
| Cifar100 | 5 | 0.8 | 0.3 | 0.1 | 1.3 | 8.0 | 0.75 |
| ImageNet100 | 7 | 0.8 | 0.3 | 0.1 | 1.5 | 8.0 | 0.75 |

## H.1 ANALYSIS OF PROTOTYPES REFINEMENT STRATEGY EPOCHS

To clarify the impact of the Prototype Refinement strategy (PR epochs), while keeping other hyper-parameters fixed, we present the results on Cifar10 dataset in Figure 9. The evaluation is based on both the overall clustering accuracy and the accuracy of pseudo labels. It's worth noting a positive correlation between overall accuracy and pseudo-label accuracy, which can be attributed to the inherent characteristics of prototypical contrastive learning and cross-entropy loss with pseudo-labels. You can find the specific refinement epochs for all datasets in table 6.

## H.2 OTHER IMPORTANT HYPER-PARAMETERS

In Figure 10, we present a sensitivity analysis of crucial hyperparameters using the Cifar10 dataset. The performance comparisons are displayed in a line plot while maintaining the other hyperparameters constant. We focus first on the loss $\mathcal{L}_{\text{SEPNet}}$. Subplots 1 to 5 (from left to right) show the impact of five weights: $\lambda_{\text{PEB}}$, $\lambda_{\text{PCL}}$, $\lambda_{\text{CE}}$, $\lambda_{\text{uni}}$, and $\lambda_l$ within the context of the loss $\mathcal{L}_{\text{SEPNet}}$. Subplot 6 examines the choice of the quantile $q$ for the Prototype Refinement strategy, where the y-axis represents the estimation error of the number of private-known categories. Our validation strategy effectively identifies the optimal values for $\lambda$PEB, $\lambda_{\text{PCL}}$, $\lambda_{\text{CE}}$, $\lambda_{\text{uni}}$, $\lambda_l$, and $q$.

