# OpenReview forum: "Semantic-Enhanced Prototypical Network for Universal Novel Category Discovery"
_ICLR.cc/2024/Conference — ICLR 2024 Conference Withdrawn Submission_

### Official Review · Reviewer_r29F · 2023-10-31

**Soundness:** 3 good
**Presentation:** 3 good
**Contribution:** 3 good
**Rating:** 5
**Confidence:** 3

**Summary:**

This paper aims to address Universal Novel Category Discovery (UniNCD) in image classification. In the UniNCD setting, the dataset categories are practically divided into three groups: "private known," "common," and "novel." Specifically, the authors have introduced a Prototype Refinement (PR) strategy, which is used during initial training to prevent negative transfer. Recognizing that finer-grained patches may contain more semantic information, the authors have associated prototypical distributions with each patch of an image. To enhance distinctiveness, the authors have introduced the patch-entropy balance loss to optimize the model. Extensive experiments demonstrate the effectiveness of this approach.

**Strengths:**

1.	In comparison to previous settings, the UniNCD setting considered in this paper is more realistic. To tackle this challenge, the authors have introduced a novel method called SEPNet. The experimental results demonstrate the effectiveness of their method.

2.	The writing and supplementary materials of this paper are well-constructed, enhancing the readers' understanding.

3.	The utilization of patch-level information to enhance semantic knowledge in contrastive learning is both interesting and effective. I believe this approach can serve as inspiration for future research.

**Weaknesses:**

1.	To balance the trade-off between smaller and larger patches, this paper employs a 2D pooling operation on the patches. However, it's worth noting that such a pooling operation may result in the loss of finer-grained information. To further investigate this trade-off, I think it is necessary to conduct corresponding experiments.

2.	Treating an image as a composition of patches is a common operation in computer vision tasks [1-2]. To enhance the novelty of this paper, further modification strategies should be applied. I think that certain filtering strategies could be effective in such task, particularly for addressing various meaningless patches (e.g., the background).

3.	The basic idea of PEB is that a concept word can be utilized to describe one image. However, what I find interesting is whether PEB can be meaningful in a multi-classification task. Further experiments could be conducted in such a scenario to explore the effectiveness of PEB.

4.	The pre-trained parameters could potentially impact the experimental results. To provide a more comprehensive illustration of the effectiveness of the PR strategy, I recommend comparing PR with other methods in the experiments.

5.	Based on the Z-score, PR distinguishes the misaligned prototypes from others. However, it may make erroneous decisions when dealing with intra-class diversity and inter-class similarity. Regarding these difficult samples, experiments should be conducted to further illustrate the effectiveness of PR.

6.	The overview in Figure 3 should be revised to include additional details about your pipeline. I found it confusing to understand the descriptions of PCL and PEB in this figure. Furthermore, it is advisable to carefully review the symbols (e.g., C_n) used in this paper to minimize unnecessary comprehension difficulties for the reader.

[1] Liu, Ze, et al. "Swin transformer: Hierarchical vision transformer using shifted windows." Proceedings of the IEEE/CVF international conference on computer vision. 2021.
[2] Dosovitskiy, Alexey, et al. "An image is worth 16x16 words: Transformers for image recognition at scale." arXiv preprint arXiv:2010.11929 (2020).

**Questions:**

See the above.

---

### Official Review · Reviewer_Hwey · 2023-11-01

**Soundness:** 3 good
**Presentation:** 2 fair
**Contribution:** 2 fair
**Rating:** 5
**Confidence:** 4

**Summary:**

The paper proposes a prototypical network (SEPNet) for the task of universal novel category discovery, which has a more practical setting compared to the standard NCD problem. The proposed SEPNet finds the optimal match between labeled and unlabeled datasets, and distinguishes the private and common known classes by utilizing a threshold based on matched prototypes' distances. Furthermore, the paper also develops a patch-entropy balance loss to promote sparsity in patch-level prototype distributions and maintains the uniformity of dense ones. The proposed method is evaluated on three benchmarks with comparison to existing approaches.

**Strengths:**

1. This paper introduces a new setting of Novel Category Discovery, which aims to address a more practical and challenging problem.
2. The proposed semantics enrichment strategy seems to be novel for the novel class representation learning.
3. Extensive experiments demonstrate the method's effectiveness, showcasing a significant performance gap compared to existing approaches.

**Weaknesses:**

1. The novelty of the proposed method is limited. The proposed method can be decoupled into two parts: 1. Prototypes Alignment. Apart from filtering private-known classes, the rest of this part is very similar to [1]. It is unclear how the proposed method differs from [1] in prototype alignment. 2. Patch-Entropy Balance for Semantic Enrichment. As explained below, $\mathcal{L}_{PEB}$ is not specifically designed for UniNCD, and using unsupervised learning loss in NCD is not new.

2. The motivation for introducing two parts is unclear. First, the Patch-Entropy Balance for the Semantic Enrichment module appears less relevant to the UniNCD task. In particular, the proposed loss
can be applied to any representation learning problem.  Second, the ablation shows that without $\mathcal{L}_{PEB}$, the SEPNet is worse than many other methods like PromptCAL and DPN w/PR.  It seems to indicate that the Prototype Refinement is less necessary for the overall framework.

3. The robustness of the threshold hyperparameter is unclear. What if the split ratio between private-known classes and common-known classes is not 3:1? It is more convincing to provide some experiments when the split ratios changes for validating the effectiveness of the model.

4. In traditional NCD & GCD tasks, there are many experiments conducted on fine-grained datasets such as Standford Cars, CUB-200-2011 and FGVC-Aircraft. Can the authors also provide similar experiment results on these fine-grained datasets?

5. The paper lacks clarity in multiple places:
 - What is the cluster label $c^u_{i,j}$ in Eq.6b?
 - What does $U_c$ represent in Eq.7?
 - The meaning of $x^j_i$ is unclear in Eq.8.
 - There are several typos in Eq.12, Eq.14, and Eq.16.
 - What is the definition of known category ratio in Appendix E? Does the ratio between private-known classes and common-known classes keep constant while the known category ratio changes? Moreover, what does "PARSE" represent in Fig 7?


[1] Wenbin An, Feng Tian, Qinghua Zheng, Wei Ding, QianYing Wang, and Ping Chen. Generalized category discovery with decoupled prototypical network. In Proceedings of the AAAI Conference on Artificial Intelligence, number 11, pp. 12527–12535, 2023.

**Questions:**

See detailed comments in Weaknesses section.

---

### Official Review · Reviewer_ASnA · 2023-11-01

**Soundness:** 3 good
**Presentation:** 2 fair
**Contribution:** 2 fair
**Rating:** 5
**Confidence:** 4

**Summary:**

This paper introduces a new task, namely Universal Novel Category Discovery (UniNCD), which involves the concept of private-known classes that is ignored by existing NCD methods. To address the negative transfer issue caused by these private-known classes, the authors proposed a prototypical contrastive learning based method with refined prototypes and enriched semantics. Extensive experiments have demonstrated the effectiveness of the proposed method.

**Strengths:**

- The proposed method is novel and evaluated to be highly effective.
- The experiments are comprehensive.

**Weaknesses:**

- The motivation of the new task seems unclear to me. For example, why holding out some private known classes is significant in the real world? Also, what does this "universal" mean?
- The computational requirement of the proposed method seems to be much more demanding than current SOTAs. For example, as another ViT-based approach, GCD only needs to train the last block of the ViT, but training the whole ViT model seems a necessity for the proposed method, if I understand correctly.
- The writing is somewhat hard to follow. There are many standalone modules and their own acronyms, making the paper not easy to understand. Also, the proposed method seems complicated, and the narration of the methodology is in lack of high-level intuitions and justification of the design choices.

**Questions:**

The proposed PEB seems to be the largest contribution. What is its intuition? Why balanced patch distribution is good? Also, in Eq 7 there is already an entropy loss term that helps rectify P towards the uniform distribution, then why another KLDU is necessary, and vice versa? If they have different impacts, apart from the theoretical analysis, is there any empirical proof?

---

### Public Comment · ~Bingchen_Zhao1 · 2023-11-12
**Questions about the problem setting and baselines**

Dear authors:

Your paper is interesting to me and relevant to my research (it cites some of my work on GCD), and after reading it, I have a few questions.

1. The difference between UniNCD and UniDA is that in UniNCD, there are no domain gaps between the labeled data and unlabeled data, thus I think by treating some of the labeled data as unlabeled, the issue of "private-known" categories can be solved, and the setting will become just GCD(or Open-World NCD as in the paper). I wonder why this is not an obvious baseline in the paper? Specifically, just duplicate the image in the labeled set and add them to the unlabeled set, and then run GCD baselines?
2. Our work [1] provides a baseline for the task of GCD, and it shows a rather strong performance across a range of datasets, I would be thankful if this paper could provide comparison to [1].
3. In GCD, evaluation datasets not only includes CIFAR and ImageNet, but also the challenge SSB benchmark [2].

Again, I'd like to say your paper is interesting, hence I raise these questions.

I understand you must have a busy schedule, but I'd like to hear your opinions on this.

Thanks

Best,

Bingchen Zhao

[1] Parametric Classification for Generalized Category Discovery: A Baseline Study, ICCV 2023.

[2] Generalized category discovery, CVPR 2022.